# pT1 Colorectal Cancer Detected in a Colorectal Cancer Mass Screening Program: Treatment and Factors Associated with Residual and Extraluminal Disease

**DOI:** 10.3390/cancers12092530

**Published:** 2020-09-06

**Authors:** Joaquín Cubiella, Antía González, Raquel Almazán, Elena Rodríguez-Camacho, Juana Fontenla Rodiles, Carmen Domínguez Ferreiro, Coral Tejido Sandoval, Cristina Sánchez Gómez, Natalia de Vicente Bielza, Isabel Peña-Rey Lorenzo, Raquel Zubizarreta

**Affiliations:** 1Department of Gastroenterology, Hospital Universitario de Ourense, Rúa Ramón Puga 52-56, 32003 Ourense, Spain; Coral.Tejido.Sandoval@sergas.es (C.T.S.); Cristina.Alejandra.Sanchez.Gomez@sergas.es (C.S.G.); Natalia.de.Vicente.Bielza@sergas.es (N.d.V.B.); 2Instituto de Investigación Sanitaria Galicia Sur, 36312 Ourense, Spain; 3Centro de Investigación Biomédica en Red Enfermedades Hepáticas y Digestivas, 32003 Ourense, Spain; 4Department of Preventive Medicine, Hospital Universitario de Ourense, 32003 Ourense, Spain; ANTIA.GONZALEZ.VAZQUEZ@sergas.es; 5Dirección Xeral de Saúde Pública, Conselleria de Sanidade, 15781 Santiago de Compostela, Spain; RAQUEL.ALMAZAN.ORTEGA@sergas.es (R.A.); Elena.Rodriguez.Camacho@sergas.es (E.R.-C.); Juana.Fontenla.Rodiles@sergas.es (J.F.R.); maria.carmen.dominguez.ferreiro@sergas.es (C.D.F.); Isabel.Pena-Rey.Lorenzo@sergas.es (I.P.-R.L.); Raquel.Zubizarreta.Alberdi@sergas.es (R.Z.)

**Keywords:** colorectal cancer, screening, pT1, endoscopic resection, overtreatment, side effects

## Abstract

**Simple Summary:**

Our study has evaluated the burden of pT1 CRC (confined to submucosa) detected during the first round of a CRC screening program, the surgery related complications and the factors related to four relevant outcomes: initial endoscopic resection, surgery rescue and residual disease after endoscopic resection and, finally, extraluminal disease after surgical resection. 38% of the CRC were detected in this stage.74.9% were initially resected endoscopically and 43.8% did not require surgery. There were inhospital surgical complications in 30.7%, mainly mild with no death and complications after discharge in 16.3% of the patients Residual disease was detected in 12 (4.3%) after endoscopic resection and extraluminal disease in 18 (8.6%) patients after surgery. We have determined several variables independently associated with the four outcome variables evaluated.

**Abstract:**

The aim of this study is to describe the treatment of pT1 colorectal cancer (CRC) in a mass screening program, the surgery-related complications and the factors associated with residual disease after endoscopic resection and extraluminal disease after surgery. We included in this retrospective analysis all the pT1 CRC detected in the Galician CRC screening program between May 2013 and June 2019. We determined which variables were independently associated with the outcomes of the study through a multivariable logistic regression analysis. We included 370–354 pT1 N0(X), 16 pT1N1- out of the 971 CRC detected; 277 (74.9%) were resected endoscopically and 162 (43.8%) were not referred to surgery. There were surgical complications in 30.7% and 16.3% of the patients during hospitalization and after discharge. Residual disease was detected in 12 (4.3%) after endoscopic resection and extraluminal disease in 18 (8.6%) patients after surgery. The variables independently associated with initial endoscopic resection were a pedunculated morphology (OR 33.1, 95% CI 4.3–254), a diameter ≥ 20 mm (OR 3.94, 95% CI 1.39–11.18) and a Site–Morphology–Size–Access score < 9 (OR 428, 95% CI 42–4263). The variables related with surgery rescue were a piecemeal resection (OR 4.48, 95% CI 1.48–13.6), an infiltrated/nonevaluable resection border (OR 7.44, 95% CI 2.12–26.0), a non-well-differentiated histology (OR 4.76, 95% CI 1.07–20.0), vascular infiltration (OR 8.24, 95% CI 2.72–25.0) and a Haggitt 4 infiltration of the submucosa (OR 5.68, 95% CI 2.62–12.3). Residual disease after endoscopic resection was associated with an infiltrated/nonevaluable resection border (OR 34.9, 95% CI 4.08–298), a non-well-differentiated histology (OR 6.67, 95% CI 1.05–50.0), and the vascular infiltration of the submucosa (OR 7.61, 95% CI 1.55–37.4). The variables related with extraluminal disease after surgical resection were no endoscopic resection (OR 4.34, 95% CI 1.26–14.28), a non-well-differentiated histology (OR 4.35, 95% CI 1.39–14.29) and the lymphatic infiltration of the submucosa (OR 4.8, 95% CI 1.32–17.8). In a CRC screening program, although most of pT1 CRC are candidates for endoscopic treatment, surgery is a safe procedure. We have defined some easy to evaluate variables that can be used in the decision-making process.

## 1. Introduction

Colorectal cancer is one of the main health problems in the Western world. In 2018, almost half a million new cases were diagnosed in Europe and 250,000 patients died due to CRC [1]. In order to reduce the burden of the disease, colorectal cancer mass screening programs have been established. This strategy has demonstrated its efficacy to reduce CRC mortality and incidence in randomized controlled trials [2,3,4]. Even more, the implementation of CRC screening programs has achieved the expected efficiency reducing both CRC mortality and incidence [5,6].

In mass screening programs, the benefit gained by individuals should outweigh any harms, for example from overdiagnosis, overtreatment, false positives, false reassurance, uncertain findings and complications [7]. Although the diagnostic performance of tests as well as related complications in CRC screening are well determined [2,3,4], there is no such certainty regarding overdiagnosis and overtreatment. In the case of CRC screening, its benefit is produced mainly to the detection of the premalignant lesions, mainly advanced complex polyps that are treated endoscopically in up to 90% of the patients [8] and the early detection of CRC. In the case of CRC, 50% is detected in stage I with an overall survival superior to 90% [9]. Furthermore, patients with a CRC confined to the submucosa (pT1) are candidates for endoscopic resection without subsequent surgery. In this case, decision-making is a balance between the risks associated with colorectal surgery and the risk of residual disease or lymph node involvement after endoscopic resection [10].

There is little information regarding the burden of pT1 CRC in a mass screening program, the treatments performed and the final outcomes, mainly associated complications and persistence or recurrence of CRC. So, we decided to perform a retrospective analysis in the Galician (northwestern Spain) CRC screening program in order to describe the treatments performed (endoscopic resection, surgery), the surgery-associated complications and the risk of residual disease after endoscopic treatment and extraluminal disease after surgical resection of pT1 CRC. Furthermore, we determined which factors were associated with four outcomes: initial endoscopic resection, surgical rescue after endoscopic resection and the presence of residual disease and/or extraluminal disease. 

## 2. Patients and Methods

### 2.1. Study Design

We designed a population-based retrospective study using the Galician (Northwestern Spain) CRC screening program database to identify the patients. We included in this analysis all the patients with a pT1 CRC detected from its implementation (May 2013) until June 2019. 

### 2.2. Description of the Galician CRC Screening Program

The Galician CRC mass screening is based on a biennial fecal immunochemical test (FIT) with 20 µg hemoglobin/g of feces threshold. FIT is offered to subjects aged between 50 and 69 years Until July 2019, 721,349 subjects were invited to participate in the screening program. The program was started in the Health Area of Ferrol in 2013, Ourense in 2015, Pontevedra, Santiago and Lugo in 2016 and A Coruña and Vigo in 2017. The mass screening program is coordinated by the Public Health Department of the Conselleria de Sanidade. They are in charge of the identification of the subjects, invitation to participate, reception of the FIT results, citation of the patients with a positive result to perform a colonoscopy and the final evaluation of the endoscopic and histological results. The main difference of the Galician’s program related to other programs existing in other regions is the coordination at the central level of the follow-up of patients depending on their risk following the EU guidelines recommendations [11]. The primary healthcare clinics are in charge of collecting FIT kits and the evaluation of the subjects with a positive FIT before colonoscopy. The hospitals in each health area are responsible for the FIT analysis, the colonoscopies, the histological analysis and the evaluation and treatment of patients with a CRC. Finally, personal at the Coordination Unit introduces data in the information system of the screening program regarding the CRC stage according to the AJCC classification [12], the final classification of the patients with a positive result [11] as well as several quality endoscopists indicators according to the Spanish guideline on quality in screening colonoscopy [13]. During the first round, the participation rate and the number of FIT positives in the first round were 42% and 6.63%, respectively.

### 2.3. Baseline Data

From each patient, we collected the information available in the screening program database: sex, age, fecal hemoglobin concentration, performance status, associated medical illnesses graded according to the American Society of Anesthesiologists’ Physical Status Classification (ASA grade) and the number of baseline colonoscopies. We collected data regarding the center and the endoscopist that performed the first complete colonoscopy. The adenoma detection rate (ADR) and the number of colonoscopies performed by each endoscopist in the first round were retrieved from the information system. We classified endoscopists in quartiles according to their ADR and the number of colonoscopies performed. Finally, we categorized hospitals according to the complexity level (tertiary versus secondary) and if they were gastroenterology teaching hospitals.

### 2.4. Treatment and Histology

After identifying the patients in the screening information system we collected data regarding the CRC from the clinical records: endoscopic evaluation and treatment, histological findings, surgery and outcomes. We determined the size, location and morphology according to the Paris classification [14]. For the analysis, lesions were classified in pedunculated and nonpedunculated, < and ≥20 mm in size and distal vs. proximal to splenic flexure On the basis of the endoscopic reports we calculated the Size, Morphology, Site and Access (SMSA) score and we classified the lesions accordingly [15]. We collected the information regarding the endoscopic resection and how it was performed (incomplete, piecemeal complete, in block complete).

With respect to surgery, we retrieved the following data: surgical approach, type of surgery, length of hospital stay and complications either during hospitalization or after discharge. We used the Clavien-Dindo classification [16] to grade the in-hospital complications. We classified surgery complications as minor if they were grade I-II and major if they were grade III–V.

Finally, in the histological evaluation, either after endoscopic or surgical resection, we determined the grade of differentiation, the lymphovascular infiltration and if tumor budding was present. In endoscopic resections, we also collected information regarding the infiltration of the resection border and the invasion depth using the Haggitt classification. We defined a high-risk pT1 CRC after endoscopic resection if any of the abovementioned conditions were met: piecemeal resection, infiltrated or nonevaluable resection border, lymphovascular infiltration, tumor budding or poor differentiation [12]. In surgical resections, we determined if there was residual lesion after endoscopic resection and the lymph node involvement.

### 2.5. Outcomes

For the analysis, we defined four outcomes.

Endoscopic resection: We determined that endoscopic resection was achieved if the resection was considered complete by the endoscopist, either piecemeal or in block.Surgical rescue: We defined surgical rescue as when surgical resection was indicated after an initial complete endoscopic resection.Residual disease after endoscopic resection: We define it as residual adenocarcinoma in the intestinal wall, in the lymph node or a relapse during follow-up.Extraluminal disease after surgical resection: We define it as a lymph node involvement detected in the surgical specimen or a relapse during follow-up.

### 2.6. Statistical Analysis

With respect to the statistical analysis, first of all, we performed a descriptive analysis of the subjects included. We described quantitative variables as the median and interquartile range (IQR) and qualitative variables as total number and percentage. Thereafter, we performed a univariate analysis using the Chi-square test and the Cochran–Mantel–Haenszel statistic (univariate logistic regression in case of a polytomous variable) to determine which variables were related to the predefined outcomes. We performed a multicollinearity analysis using the variable inflation factor analysis to exclude collinearity in the variables statistically significant. Finally, we include these variables in a multivariable analysis using logistic regression (forward conditional) to determine those independently related to the outcomes. Differences were expressed as odds ratio (OR) with a 95% confidence interval (CI). Statistical analyses were performed with IBM SPSS Statistics for Windows, Version 22.0. Armonk, NY: IBM Corp.

### 2.7. Ethical Issues

The study was conducted in accordance with the Declaration of Helsinki, and the protocol was approved by the Ethics Committee of Galicia, Spain (code 2018/593). As long as the study was based on database exploitation, no informed consent was required. The access to the information was performed according to the European and Spanish legislation. 

## 3. Results

During the study period, 971 invasive CRC were detected: 482 (49.6%) TNM I, 186 (19.1%) TNM II, 246 (25.3%) TNM III and 57 (5.9%) TNM IV. We included in this analysis 370 patients that met the inclusion criteria: 354 pT1N0(X), 73.4% of the TNM I; and 16 pT1N1, 6.5% of the TNM III (Figure 1). They were mainly male (69.1%), older than 60 years (65.4%) and required only a baseline colonoscopy for the diagnosis and/or treatment in 85.1% of the cases. With respect to the detected lesion, they were predominantly located distal to the splenic flexure (87.8%). The most frequent morphology was pedunculated (47.6%), the median size was 18 mm (IQR 12–25 mm) and the median SMSA score was eight (IQR 6–11). 

Seventy-one endoscopists from seven hospitals participated in the first round of the CRC screening program. The median number of colonoscopies performed was 278 (IQR 56–507) and the median ADR was 65.3% (IQR 60.0–70.1%). According to the complexity, hospitals were classified in tertiary (3) and secondary (4). Six of the hospitals were gastroenterology teaching hospitals when CRC screening was implemented.

### 3.1. Endoscopic Resection

An endoscopic resection was attempted in 283 (76.5%) patients: incomplete in 6 (1.6%), piecemeal complete in 55 (14.9%) and in block complete in 222 (60.0%). All patients with incomplete or no endoscopic resection were referred to surgery. In the univariate analysis (Table 1) several factors were related to a complete endoscopic resection, mainly related to anesthesiology risk, the characteristics of the lesion (location, size, morphology and the classification according to the SMSA score) and the level of complexity of the hospital. In the seven endoscopy units participating in the CRC screening program, the rate of initial endoscopic resection ranged between 46.7% and 83.7% (*p* = 0.01). However, only the pedunculated morphology (OR 33.1, 95% CI 4.3–254), a diameter larger than 20 mm (OR 3.94, 95% CI 1.39–11.18) and an SMSA score below 9 (OR 427, 95% CI 42–4263) were independently related to a complete endoscopic resection (Figure 2).

### 3.2. Surgery Rescue and Residual Disease after Endoscopic Resection

After a complete endoscopic resection, 115 (41.5%) patients required surgical resection (Figure 1). In the seven hospitals, the rate of surgery after the endoscopic resection ranged between 14.3% and 56.5% (*p* = 0.1). We identified several factors related to the lesion, the endoscopic resection and the histological diagnosis associated with the surgical rescue in the univariate analysis. However, only a piecemeal resection (OR 4.48, 95% CI 1.48–13.6), an infiltrated/nonevaluable resection border (OR 7.44, 95% CI 2.12–26.0), a non-well-differentiated histology (OR 4.76, 95% CI 1.07–20.0), vascular infiltration (OR 8.24, 95% CI 2.72–25.0) and a Haggitt 4 infiltration of the submucosa (OR 5.68, 95% CI 2.62–12.3) were independently related to the referral to surgery (Table 2, Figure 2).

After the endoscopic resection, the patients had a median follow-up of 24.4 months (IQR 17.2–31.8). During this period, nine (3.2%) died. No recurrence was detected in the patients with an initial endoscopic resection. In the patients that required surgery, a residual disease in the intestinal wall was detected in seven (6.1%) and lymph node involvement in five (4.3%) patients with an overall rate of residual disease of 4.3% (Figure 1). The risk of residual disease in the patients that underwent surgery was 10.4%. As we show in Table 3, only three variables were independently related to the risk of residual disease after endoscopic resection: an infiltrated/nonevaluable resection border (OR 34.9, 95% CI 4.08–298), a non-well-differentiated histology (OR 6.67, 95% CI 1.05–50.0) and the vascular infiltration of the submucosa (OR 7.61, 95% CI 1.55–37.4) (Figure 2).

### 3.3. Surgery and Related Complications

In the seven hospitals participating in the screening program, the global surgery rate ranged between 45.5% and 72.6% (*p* = 0.05). The most frequent surgical approach was laparoscopy accounting for 75.5% of the surgeries. With respect to the type of surgery, the most frequent were sigmoidectomy and anterior resection of the rectum, as we show in Table 4. The median hospitalization was seven days and complications were detected in 30.7% of the patients, minor in 24% and major in 6.7%. There was no death during hospitalization. During a median follow-up of 25.6 months, 11 patients (3.0%) died and surgically related complications were detected in 34 patients (16.3%), mainly abdominal reinterventions (11), anastomotic stenosis (5) and intestinal occlusion (5). Finally, an initial endoscopic resection did not modify the complications rate. The risk of in-hospital (OR 0.9, 95% CI 0.50–1.71) or after discharge (OR 0.58; 95% CI 0.28–1.22) complications were similar between both groups.

### 3.4. Extraluminal Disease after Surgery Resection

In the 208 patients referred to surgery, a lymph node involvement was detected in 16 (7.7%) patients and a distant recurrence in two (1.9%), with an overall rate of extraluminal disease after surgical resection of 8.6% (Figure 1). We identified three factors independently related with extraluminal disease: no endoscopic resection (OR 4.34, 95% CI 1.26–14.28), a non-well-differentiated histology (OR 4.35, 95% CI 1.39–14.29) and lymphatic infiltration of the submucosa (OR 4.8, 95% CI 1.32–17.8) as we display in Table 5 and Figure 2.

## 4. Discussion

Our study shows the relevance of pT1 CRC in a mass screening program. Up to 38% of the CRC are detected in this stage and, thus, are potential candidates for endoscopic treatment. Although initially three-quarters of the lesions were resected endoscopically, more than half of the patients finally underwent colorectal surgery. Our analysis has evaluated which factors were associated with the relevant outcomes in the treatment of pT1 CRC: endoscopic resection, surgery rescue, residual disease after endoscopic resection and extraluminal disease after surgery.

In the treatment of pT1 CRC, we have to weigh the benefits and risks of the two treatment options available: surgical and endoscopic resection [17]. Endoscopic resection has few side effects, complications not superior to 1–2% and mortality below 1/10,000 [2,3,4]. On the other hand, the main risk is related to the intestinal wall residual disease after incomplete resections and the lymph node involvement [10,18]. Moreover, surgery allows resecting completely and, thus, evaluates both the colon wall and the regional lymph nodes. Although it is associated with a low mortality risk, especially in the laparoscopic approach [19], it produces relevant short- and long-term complications [20]. Traditionally, in the decision-making analysis, the risk of death after surgery has been confronted with the risk of residual disease after endoscopic resection [17]. In this sense, several variables, mainly related to the endoscopic resection and the histological analysis, allows us to determine a low and high-risk group for residual disease after endoscopic resection [18,21,22,23,24]. Nevertheless, the risk of surgical related long-term morbidity is usually not taken into consideration. 

In the centers participating in the Galician CRC screening program, the criteria used to refer patients to surgery were very specific as long as no recurrence was detected after a two years median surveillance. On the other hand, on account of the mortality and the risk of residual disease; in the group that underwent surgical rescue, the number of patients to detect a residual disease was 9.6. Nevertheless, our results confirm the discrepancies between the available recommendations to stratify patients in the low- and high-risk group and the criteria used to refer to surgery [25]. In our case, 23.1% of the low-risk patients were referred to surgery and, on the other hand, 32% of the patients in the high-risk group were kept in surveillance. These discrepancies may be related to the evaluation and interpretation of the resection border: piecemeal resections and distance to the border.

Our study has several strengths. The first one is that we have assessed which factors are associated with an initial endoscopic resection. In this sense, we have evaluated the SMSA classification for the first time in the pT1 CRC confirming that it fairly discriminates which lesions are candidates for endoscopic resection. SMSA was first described by Gupta et al. [26] and has confirmed in several studies its ability to identify which lesions are challenging [27,28]. In this sense, it is relevant to improve the resection skills of the endoscopists in order to evaluate which lesions are at risk of harboring an invasive CRC candidate for endoscopic resection and, in any case, obtain a complete endoscopic resection according to the characteristics of the lesion. It is important to remind that endoscopic resection must enable the evaluation of the resection border [19,29]. As we have shown in this analysis, this is one of the discriminant variables associated with the risk of residual disease after endoscopic resection.

One of the dilemmas the endoscopists have when they suspect a lesion contains an invasive CRC is if it should be resected or left for surgical resection. The first limitation is that the available visual predicted classifications have a limited specificity for invasive CRC. As an example, the Narrow-Band Imaging International Colorectal Endoscopic (NICE) classification sensitivity for invasive CRC only reaches 58% in a recently published study [30]. Furthermore, the positive predictive value of the NICE classification for invasive carcinoma is only above 50% in the depressed NICE III lesions. On the other hand, there are doubts if an initial endoscopic resection could increase the associated risks or difficult a posterior surgery. Our results confirm that an initial endoscopic resection has no effect on the complications rate [31]. Furthermore, an initial endoscopic resection is independently associated with a reduced risk of extraluminal disease. So, we suggest that a complete endoscopic resection should always be attempted as long as we will avoid unnecessary surgeries (benign lesions and low-risk pT1 CRC) and it will allow us to stratify the risk of extraluminal disease in case surgery is finally required with no effect on the final outcomes.

Our study has several limitations, mostly related to its retrospective nature. We are lacking uniformity in the histological evaluation among centers and there are several relevant variables that are lacking, mainly tumor budding and the depth of invasion using the Kikuchi scale [18,21,22,23,24,25,32]. This last is the most relevant as long as there is a direct relation between submucosa depth of invasion and the risk of residual disease and lymph node involvement. Our results are concordant with other studies that evaluate the real practice in pT1 CRC [33]. In this sense, our study highlights the need to standardize the evaluation of pT1 CRC in polypectomies specimens and to establish a continuous quality improvement policy in the pathology departments [34]. Besides, we do not have information regarding the visual classification according to any of the available scores, mainly NICE. Probably, this information would be relevant to determine the sensitivity in this setting, the FIT-based CRC mass screening program, and if it correlates with any of the defined outcomes.

Our study shows there are several areas of improvement for the future. Piecemeal resections should be reduced. In our study, nearly 20% of the resections were fragmented, thus increasing the risk of residual disease and restraining the evaluation of the resection border. Although the visual evaluation and the endoscopic resection techniques should be improved, we have to draw attention to the high ADR in the endoscopists participating in the Galician screening program. Although an ADR higher than 45% is recommended in FIT-based screening programs [35], in our case, 75% of the endoscopists reached a 60% ADR. So, endoscopic resection techniques such as submucosal dissection and endoscopic full-thickness resection should be available and patients should be referred to centralized units where these techniques are performed on a regular basis [34]. On the other hand, as we have stated previously, a standardized histological evaluation is mandatory. Nevertheless, we require more accurate histological criteria to predict the risk of lymph node involvement and, thus, avoid unnecessary surgeries in patients with a negligible positive predictive value for local node infiltration. Finally, the laparoscopic–endoscopic cooperative surgery in pT1 CRC together with the colonoscopy tattooing and the sentinel lymph node mapping could be an option to reduce both the short- and long-term complications associated with colectomy in the high-risk patients after complete endoscopic resection [36,37,38].

## 5. Conclusions

To conclude, pT1 CRC are a high proportion of the CRC detected in a mass screening program. The risk of residual disease or relapse in the low-risk group after endoscopic resection and of mortality in patients undergoing surgery was zero. However, we need to improve the endoscopic resection techniques, the histological evaluation and to evaluate new hybrid endoscopic and surgical approaches to reduce the burden of the treatment with the same oncological results.

## Figures and Tables

**Figure 1 cancers-12-02530-f001:**
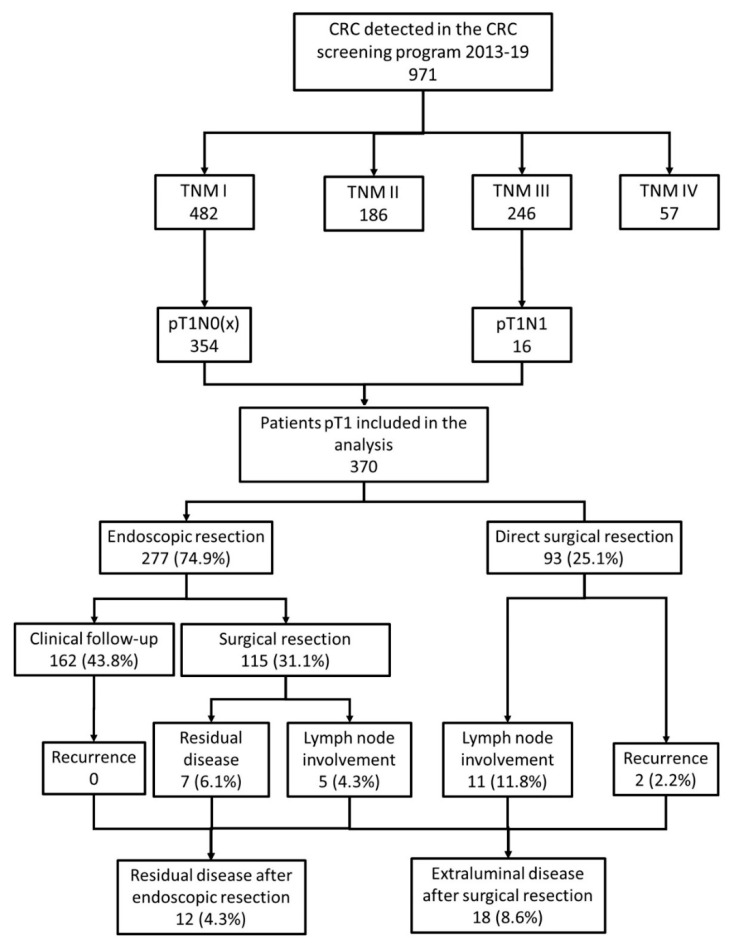
Flowchart of the study. We show the inclusion in the analysis and the main outcomes of the study: endoscopic resection, surgery, residual disease after endoscopic resection and extraluminal disease after surgical resection.

**Figure 2 cancers-12-02530-f002:**
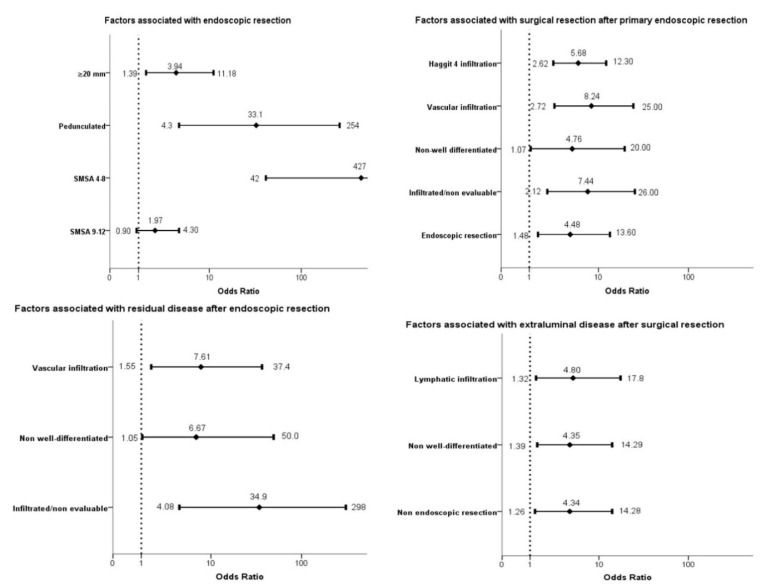
Forest plot. We show the variables independently associated with the four outcomes in the multivariable logistic regression analysis (forward conditional). The association is expressed as odds ratio and its 95% confidence interval.

**Table 1 cancers-12-02530-t001:** Factor associated with endoscopic resection.

Variables Evaluated		Endoscopic Resection (*n* = 277)	Direct Surgery (*n* = 93)	Odd Ratio (95% CI) ^1^	Odd Ratio (95% CI) ^2^
Gender (*n* = 370)	Female Male	87 (76.3%) 190 (74.2%)	27 (23.7%) 66 (25.8%)	1 0.90 (0.53–1.50)	
Age (*n* = 370)	<60 years ≥60 years	97 (75.8%) 180 (74.4%)	31 (24.2%) 62 (25.6%)	1 0.93 (0.56–1.52)	
PS (*n* = 361)	0 1	236 (74.7%) 39 (84.8%)	80 (25.3%) 7 (15.2%)	1 1.84 (0.79–4.28)	
ASA (*n* = 361)	I II III	161 (71.6%) 99 (83.2%) 14 (73.7%)	64 (28.4%) 18 (16.8%) 5 (26.3%)	1 2.20 (1.22–3.90) 1.11 (0.38–3.21)	
Fecal Hb (*n* = 370)	<200 µg/g ≥200 µg/g	178 (74.2%) 99 (76.2%)	62 (25.8%) 31 (23.8%)	1 1.11 (0.68–1.82)	
N. colonoscopies (*n* = 346)	One At least two	242 (76.8%) 26 (83.9%)	73 (23.2%) 5 (16.1%)	1 1.56 (0.58–4.23)	
Lesion size (*n* = 370)	<20 mm ≥20 mm	160 (85.6%) 117 (63.9%)	27 (14.4%) 66 (36.1%)	1 0.30 (0.18–0.50)	1 3.94 (1.39–11.18)
Morphology (*n* = 370)	Nonpedunculated Pedunculated	103 (53.1%) 174 (98.9%)	91 (46.9%) 2 (1.1%)	1 76.8 (18.5–318.6)	1 33.1 (4.3–254)
Location (splenic flexure) (*n* = 370)	Proximal Distal	21 (46.7%) 256 (78.8%)	24 (53.3%) 69 (21.2%)	1 4.24 (2.23–8.10)	
SMSA classification (*n* = 370)	>12 9–12 4–8	34 (45.3%) 50 (49.5%) 203 (95.5%)	41 (54.7%) 51 (50.5%) 1 (0.5%)	1 1.67 (0.89–3.17) 346 (46–2636)	1 1.97 (0.90–4.30) 427 (42–4263)
Endoscopist ADR (*n* = 369)	Q1 (≤60%) Q2 (60–65.3%) Q3 (65.3–70.8%) Q4 (>70.8%)	58 (65.9%) 61 (78.2%) 71 (77.2%) 87 (78.4%)	30 (34.1%) 17 (21.8%) 21 (22.8%) 24 (21.6%)	1 1.85 (0.93–3.72) 1.75 (0.91–3.37) 1.87 (0.99–3.52)	
Endoscopist number of colonoscopies (*n* = 369)	≤278 >278	35 (81.4%) 242 (74.2%)	8 (18.6%) 84 (25.8%)	1 0.66 (0.29–1.48)	
High-risk CRC clinic (*n* = 370)	No Yes	84 (77.8%) 193 (73.7%)	24 (22.2%) 69 (26.3%)	1 0.80 (0.47–1.36)	
Complexity of hospital (*n* = 370)	Second level (*n* = 4) Third level (*n* = 3)	149 (70.3%) 128 (81.0%)	63 (29.7%) 30 (19.0%)	1 1.80 (1.10–2.96)	

^1^ Odds Ratio and 95% CI calculated in the univariate analysis using the Cochran–Mantel–Haenszel statistic. ^2^ Odds Ratio and 95% CI calculated in the multivariable logistic regression analysis (forward conditional). ADR, adenoma detection rate; ASA, American Society of Anesthesiologists’ Physical Status; CI, confidence interval; CRC, colorectal cancer; PS, performance status; SMSA, Size, Morphology, Site and Access.

**Table 2 cancers-12-02530-t002:** Factors associated with surgical resection after primary endoscopic resection.

Variables Evaluated		Clinical Follow-up (*n* = 162)	Surgical Resection (*n* = 115)	Odd Ratio (95% CI) ^1^	Odd Ratio (95% CI) ^2^
Gender (*n* = 277)	Female Male	52 (59.8%) 110 (57.9%)	35 (40.2%) 80 (42.1%)	1 1.08 (0.64–1.81)	
Age (*n* = 277)	<60 years ≥60 years	55 (56.7%) 107 (59.5%)	42 (43.3%) 73 (40.5%)	1 0.89 (0.54–1.47)	
Colonoscopies (*n* = 258)	One At least two	134 (57.8%) 21 (80.8%)	98 (42.2%) 5 (19.2%)	1 0.32 (0.12–0.89)	
Lesion size (*n* = 277)	<20 mm ≥20 mm	96 (60.0%) 66 (56.4%)	64 (40.0%) 51 (43.6%)	1 1.16 (0.72–1.88)	
Morphology (*n* = 277)	Nonpedunculated Pedunculated	36 (35.0%) 126 (72.4%)	67 (65.0%) 48 (27.6%)	1 0.20 (0.12–0.35)	
Location (splenic flexure) (*n* = 277)	Proximal Distal	7 (33.3%) 155 (60.6%)	14 (66.7%) 101 (39.4%)	1 0.33 (0.13–0.83)	
SMSA classification (*n* = 277)	4–8 9–12 >12	138 (68.0%) 21 (42.0%) 3 (12.5%)	65 (32.0%) 29 (58.0%) 21 (87.5%)	1 2.93 (1.55–5.53) 14.8 (4.3–51.6)	
Endoscopic resection (*n* = 277)	In block Piecemeal	152 (68.5%) 10 (18.2%)	70 (31.5%) 45 (81.8%)	1 9.78 (4.66–20.5)	1 4.48 (1.48–13.6)
Infiltration of the border (*n* = 277)	Noninfiltrated Infiltrated/nonevaluable	154 (71.3%) 8 (13.1%)	62 (28.7%) 53 (86.9%)	1 16.5 (7.40–36.6)	1 7.44 (2.12–26.0)
Well-differentiated (*n* = 277)	Yes No	5 (23.8%) 157 (61.4%)	16 (76.2%) 99 (38.6%)	5.00 (1.81–14.3) 1	4.76 (1.07–20.0) 1
Lymphatic infiltration (*n* = 264)	No Yes	153 (61.7%) 2 (12.5%)	95 (38.3%) 14 (87.5%)	1 11.3 (2.51–50.7)	
Vascular infiltration (*n* = 277)	No Yes	153 (67.0%) 9 (26.5%)	90 (37.0%) 25 (73.5%)	1 4.72 (2.11–10.5)	1 8.24 (2.72–25.0)
Tumor budding (*n* = 48)	No Yes	13 (61.9%) 16 (59.3%)	8 (38.1%) 11 (40.7%)	1 1.12 (0.35–3.6)	
Haggitt classification (*n* = 224)	<4 4	102 (85.6%) 34 (33.3%)	20 (16.4%) 68 (66.7%)	1 10.2 (5.42–19.2)	1 5.68 (2.62–12.3)
High-risk pT1 CRC (*n* = 277)	No Yes	130 (76.9%) 32 (29.6%)	39 (23.1%) 76 (70.4%)	1 7.92 (4.58–13.6)	
Endoscopist ADR (*n* = 277)	Q1 (≤60%) Q2 (60–65.3%) Q3 (65.3–70.8%) Q4 (>70.8%)	29 (50.0%) 36 (52.1%) 48 (67.6%) 49 (56.3%)	29 (50.0%) 25 (47.9%) 23 (32.4%) 38 (43.7%)	1 0.69 (0.34–1.43) 0.48 (0.23–0.98) 0.78 (0.40–1.51)	
Endoscopist number of colonoscopies (*n* = 277)	≤278 >278	25 (71.5%) 137 (56.6%)	10 (28.5%) 105 (43.4%)	1 1.92 (0.88–4.16)	
Complexity of hospital (*n* = 370)	Second level Third level	81 (54.4%) 81 (63.3%)	68 (45.6%) 47 (36.7%)	1 0.69 (0.42–1.12)	

^1^ Odds Ratio and 95% CI calculated in the univariate analysis using the Cochran–Mantel–Haenszel statistic. ^2^ Odds Ratio and 95% CI calculated in the multivariable logistic regression analysis (forward conditional). ADR, adenoma detection rate; CI, confidence interval; CRC, colorectal cancer; SMSA, Size, Morphology, Site and Access.

**Table 3 cancers-12-02530-t003:** Factors associated with residual disease after endoscopic resection.

Variables Evaluated		No Residual Disease (*n* = 265)	Residual Disease (*n* = 12)	Odd Ratio (95% CI) ^1^	Odds Ratio (95% CI) ^2^
Diameter (*n* = 277)	<20 mm ≥20 mm	153 (95.6%) 112 (45.8%)	7 (4.4%) 5 (4.2%)	1 0.98 (0.30–3.15)	
Morphology (*n* = 277)	Nonpedunculated Pedunculated	95 (92.8%) 170 (97.7%)	8 (7.8%) 4 (2.3%)	1 0.28 (0.08–0.95)	
Location (splenic flexure) (*n* = 277)	Proximal Distal	21 (100%) 244 (95.3%)	0 12 (4.7%)		
SMSA classification (*n* = 277)	4–8 9–12 >12	196 (96.5%) 47 (94.0%) 22 (91.7%)	7 (3.5%) 3 (6.0%) 2 (8.3%)	1 1.79 (0.44–7.17) 2.54 (0.50–13.0)	
Endoscopic resection (*n* = 277)	In block Piecemeal	214 (96.4%) 51 (92.7%)	8 (3.6%) 4 (7.3%)	1 2.10 (0.61–7.24)	
Infiltration of the border (*n* = 277)	Noninfiltrated Infiltrated/nonevaluable	213 (98.6%) 52 (85.3%)	3 (1.4%) 9 (14.7%)	1 12.3 (3.21–47.0)	1 34.9 (4.08–298)
Well-differentiated (*n* = 277)	No Yes	18 (85.7%) 247 (96.5%)	3 (14.3%) 9 (3.5%)	4.5 (1.14–20.0) 1	6.67 (1.05–50.0) 1
Lymphatic infiltration (*n* = 264)	No Yes	240 (96.8%) 14 (87.5%)	8 (3.2%) 2 (12.5%)	1 4.29 (0.83–22.1)	
Vascular infiltration (*n* = 277)	No Yes	236 (97.1%) 29 (85.3%)	7 (2.9%) 5 (14.7%)	1 5.91 (1.73–19.5)	1 7.61 (1.55–37.4)
Tumor budding (*n* = 48)	No Yes	21 (100%) 26 (78.8%)	0 7 (21.2%)		
Haggitt classification (*n* = 224)	<4 4	121 (91.2%) 93 (91.2%)	1 (0.8%) 9 (8.8%)	1 11.7 (1.46–94.1)	
High-risk pT1CRC (*n* = 277)	No Yes	168 (99.4%) 97 (89.8%)	1 (0.6%) 11 (10.2%)	1 19.0 (2.4–149)	

^1^ Odds Ratio and 95% CI calculated in the univariate analysis using the Cochran–Mantel–Haenszel statistic. ^2^ Odds Ratio and 95% CI calculated in the multivariable logistic regression analysis (forward conditional). CI, confidence interval; CRC, colorectal cancer; SMSA, Size, Morphology, Site and Access.

**Table 4 cancers-12-02530-t004:** Surgical approach and associated complications in the patients included in the study.

Surgery and Associated Complications		Number
Surgical approach (*n* = 208)	Laparoscopy Reconverted laparoscopy Laparotomy Transanal surgery	142 (68.3%) 15 (7.2%) 35 (16.8%) 16 (7.7%)
Type of surgery (*n* = 208)	Right hemicolectomy Left hemicolectomy Sigmoidectomy Rectum anterior resection Abdominoperineal resection Segmental resection Subtotal colectomy Transanal surgery	33 (15.9%) 17 (8.2%) 87 (41.8%) 45 (21.6%) 3 (1.4%) 3 (1.4%) 4 (1.9%) 16 (7.7%)
Length of hospitalization (days)		7 (IQR 6–9.75)
In-hospital complications (*n* = 208)	0 I II III IV V	154 (74.0%) 32 (10.6%) 18 (8.7%) 11 (5.3%) 3 (1.4%) 0 (0.0%)
Follow-up after discharge (months)		25.6 (18.5–35.4)
Complications after discharge (*n* = 208)		34 (16.3%)
Death		11 (3.0%)

**Table 5 cancers-12-02530-t005:** Factors associated with extraluminal disease after surgical resection.

Variables Evaluated		No Extraluminal Disease (*n* = 190)	Extraluminal Disease (*n* = 18)	Odd Ratio (95%CI) ^1^	Odds Ratio (95% CI) ^2^
Diameter (*n* = 208)	<20 mm ≥20 mm	93 (90.3%) 97 (92.4%)	10 (9.7%) 8 (7.6%)	1 0.77 (0.29–2.03)	
Morphology (*n* = 208)	Nonpedunculated Pedunculated	142 (89.9%) 48 (96.0%)	16 (10.1%) 2 (4.0%)	1 0.37 (0.08–1.67)	
Location (splenic flexure) (*n* = 208)	Proximal Distal	36 (94.7%) 154 (90.6%)	2 (5.3%) 16 (9.4%)	1 1.87 (0.41–8.5)	
SMSA classification (*n* = 208)	4–8 9–12 >12	61 (92.2%) 71 (88.8%) 58 (93.6%)	5 (7.8%) 9 (11.2%) 4 (6.4%)	1 1.55 (0.49–4.86) 0.84 (0.21–3.29)	
Endoscopic resection (*n* = 208)	No Yes	80 (86.0%) 110 (95.7%)	13 (14.0%) 5 (4.3%)	3.57 (1.22–10.0) 1	4.34 (1.26–14.28) 1
Well-differentiated (*n* = 204)	Yes No	159 (93.5%) 27 (79.4%)	11 (6.5%) 7 (20.6%)	3.70 (1.35–11.11) 1	4.35 (1.39–14.29) 1
Lymphatic infiltration (*n* = 193)	No Yes	159 (93.6%) 18 (78.3%)	11 (6.4%) 5 (21.7%)	1 4.01 (1.25–12.8)	1 4.80 (1.32–17.8)
Vascular infiltration (*n* = 199)	No Yes	152 (93.3%) 30 (83.3%)	11 (6.7%) 6 (16.7%)	1 2.76 (0.95–8.05)	
Tumor budding (*n* = 65)	No Yes	41 (87.2%) 17 (94.4%)	6 (12.8%) 1 (5.6%)	1 0.40 (0.04–3.60)	

^1^ Odds Ratio and 95% CI calculated in the univariate analysis using the Cochran–Mantel–Haenszel statistic. ^2^ Odds Ratio and 95% CI calculated in the multivariable logistic regression analysis (forward conditional). CI, confidence interval; SMSA, Size, Morphology, Site and Access.

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
