# Peer review of "pT1 Colorectal Cancer Detected in a Colorectal Cancer Mass Screening Program: Treatment and Factors Associated with Residual and Extraluminal Disease"

_cancers, 2020, doi:10.3390/cancers12092530_

Round 1

Reviewer 1 Report

It is a very interesting work with practical applicability when the endoscopic resection of pT1 disease is increasingly employed. Even with the limitations, which the authors mention in the manuscript, the observed results came to add evidence to other studies that evaluate the real practice in pT1 CRC.

The authors collect a considerable number of patients and the manuscript is well conceived, written and developed. The tables provide sufficient information.

ABSTRACT

Please rephrase “We included 370- 354 pT1 N0(X), 16 pT1N1- out of the 971 CRC detected, 74.9% were resected endoscopically and 43.8% were not referred to surgery.”

It is not obvious at first glance if “74.9%” refers to 370 or to 971.

METHODS

I would subcategorize the methods section in order to make it easier to read and follow it. I’d suggest, but it’s up to the authors, something like:

  • The Galician CRC mass screening program and Patient population
    • Lines 69 to 119
  • Study outcomes
    • Lines 120 to 129
  • Statistical considerations
    • Line 129 to 143

Although the context makes it obvious that “extraluminal disease” is related to lymph node involvement or metastatic disease, please consider clearly define it that way.

I’m afraid it is not correct to group together two completely different diseases: pT1pN0 and pT1pN1. Their prognoses are so different that they are staged separately. By Definition, pT1pN1 already has extraluminal disease. Therefore, at least for this analysis, pT1pN1 must be excluded. This is my only major concern about this manuscript.

RESULTS

Almost every time the authors cite an Odds Ratio (OR) in the text, they use ORs pointed to different directions (harm and protection). For instance, line 205: “We identified three factors independently related with 206 extraluminal disease: endoscopic resection (OR 0.23, 95% CI 0.07-0.79), well differentiated histology 207 (OR 0.23, 95% CI 0.07-0.72) and lymphatic infiltration of the submucosa (OR 4.8, 95% CI 1.32-17.8) as we display in table 5”.

Although statistically correct, it is much easier for anyone (even those not familiar with the concept of ORs) to understand if it had been written as the following:

We identified three factors independently related with extraluminal disease: not having endoscopic resection (OR 4.35, 95% CI x.xx – x.xx), non-well differentiated histology 207 (OR 4.35, 95% CI x.xx – x.xx) and lymphatic infiltration of the submucosa (OR 4.8, 95% CI 1.32-17.8) as 208 we display in table 5”.

Please understand that many physicians are not familiar with some statistical concepts, sometimes even the most basic ones. Besides, odds are fairly easy to visualize when they are greater than one, but are less easily grasped when the value is less than one.

TABLES

I believe it is possible to reshape the tables so the OR and 95%CI could be placed at the same line of its related data.

Table 5 – Line “Well differentiated”: Having “yes” as reference (OR=1), “No” had a protective effect (OR=0.27). Is that correct? Tumor budding had the same “paradoxical” effect. However, it was a completely underpowered analysis (n=65; 95%CI 0.04- 3.60), and must be considered exploratory only.

Although tables are sufficient, the information would be much better presented if illustrated in forest plots. At least for the multivariable logistic regression models. I’m not sure if is possible to be done using SPSS, but it sure is using R packages, like the following (this is just a suggestion to your statistician):

# Import data to “dat”

install.packages("readxl")

library(readxl)

dat <- read_excel("C:/your data.xlsx”)

# be sure that all variables are categorical variables, for instance:

dat$Endoscopic_resection <- as.factor(dat$Endoscopic_resection)

# Run binary logistic regression (where “Extraluminal_disease” is zero or 1)

Logistic_extraluminal <- glm(Extraluminal_disease  ~ Endoscopic_resection + Well_differentiated + Lymphatic_infiltration, data = dat, family = "binomial")

# Forest plot

install.packages("forestmodel")

library(forestmodel)

forest_model(Logistic_extraluminal)

Author Response

Please see the cover letter of the revised manuscript (reviewer 1).

Reviewer 2 Report

The authors presented the treatment of pT1 colorectal cancer (CRC) and its burden in a mass screening program in specific area, and analyzed factors associated with the outcomes, such as the surgery related complications, residual disease after endoscopic resection, and extraluminal disease after surgery.

Adding data about hospital and endoscopists from community hospitals, and using SMSA score are new aspects in this study, compared to previous studies from single or several tertiary institutes. However, most of results are similar with those of previous studies, and there is no novel findings.

You need to show something new or confirming controversial issues, comparing with previous studies. 

Author Response

Please see the cover letter of the revised manuscript (reviewer 2)

Reviewer 3 Report

In the manuscript entitled “pT1 colorectal cancer detected in a colorectal cancer mass screening program: treatment and factors associated with residual and extraluminal disease.” Cubiella et al. have carried out a retrospective analysis of the risks and complications associated with endoscopic resection and surgery in 370 patients with pT1. They have incorporated several variables that would affect these aspects including the experience of the personnel carrying out the procedure. The authors have stated the issue well in the introduction. They have neatly discussed the associations and also made some suggestions to reduce the complications. Considering the high frequency of colorectal cancer occurrence and the associated mortality this manuscript seems to be making a valuable input in improving the colorectal cancer outcome. However, there are following issues with the drafting of the manuscript that limit its readability and need to be addressed before it could be considered for publication in Cancers.

Minor corrections

  1. There are too many typographical errors and incorrect use of language throughout the manuscript.

Major concerns

  1. Introduction, discussion, and conclusion are well written but the language in the rest of the sections-abstract, methods, and results, needs to be made less complicated and easy to understand.
  2. The abstract is also missing the analysis-based conclusion of the study.
  3. The duration of the sample collection is different in the abstract and method sections- July 2019 and June 2019.
  4. Line 73, 20ug of what per g feces?
  5. The numbers of samples in the flow chart and text in the results section don’t match in certain instances.
  6. Authors have merged data from pT1N0 and pT1N1 but segregation should be maintained all throughout the subsequent analysis. Contribution from the advanced pT1N1 samples, though the number is small, may mask the significance of some parameters that would otherwise better correlate with different outcomes.

Author Response

Please see the cover letter of the revised manuscript (reviewer 3)

Round 2

Reviewer 1 Report

No more comments

Reviewer 2 Report

The authors answred all comments from reviewers.

I don't have further comments.

Reviewer 3 Report

The authors have addressed all the concerns raised in the first review. I do not have any further comments.